# Intelligent Motor Bearing Fault Diagnosis Using Channel Attention-Based CNN

**Jianguo Yin and Gang Cen ***

School of Information and Electronic Engineering, Zhejiang University of Science and Technology, Hangzhou 310023, China
* Correspondence:gcen@zust.edu.cn

**Abstract:** Many components of electric vehicles contain rolling bearings, and the operating condition of rolling bearings often affects the operating performance of electric vehicles. Monitoring the operating status of the bearings is one of the key technologies to ensure the safe operation of the bearings. We propose a channel attention-based convolutional neural network (CA-CNN) model for rolling bearing fault diagnosis. The model can directly use the raw vibration signal of the bearing as input to achieve bearing fault diagnosis under different operating loads and different noise environments. The experimental results show that, compared with other intelligent diagnosis methods, the proposed model CA-CNN achieves a high diagnostic accuracy under different load cases and still has advantages in different noisy environments. It is also beneficial to promote the intelligent fault diagnosis and maintenance of electric vehicles.

**Keywords:** motor bearing; fault diagnosis; convolutional neural networks; channel attention; deep learning

## 1. Introduction

Permanent magnet synchronous motors are the main power output components of electric vehicles [1], and bearings are the core components of drive motors, which often work in complex environments, this requires a high reliability of bearings during operation. Therefore, it is important to carry out bearing fault diagnosis methods for the timely maintenance of motors.

In recent years, fault diagnosis methods based on machine learning have received more and more attention and are widely used in the field of bearing diagnosis. Asr et al. [2] utilized empirical mode decomposition (EMD) and a non-naïve Bayesian classifier (NNBC) to effectively improve the accuracy of bearing fault diagnosis in automobile gearboxes. Jiang et al. [3] proposed an adaptive bearing fault diagnosis method based on variational mode decomposition (VMD) and multi-resolution Teager energy operator (MTEO). Ali et al. [4] used artificial neural networks (ANN) for bearing fault detection. Santos et al. [5] analyzed the power signal of a wind turbine and used a support vector machine (SVM) to troubleshoot it. Shevchik et al. [6] used signal processing methods of acoustic emission and random forests (RF) to predict the wear of bearings. Although these machine learning-based fault diagnosis methods have made some progress, they usually rely on expert knowledge and hardly meet the needs of complex industrial environments [7].

With the development of deep learning, deep neural network (DNN)-based methods have driven the research in the field of fault diagnosis. The convolutional neural network (CNN), as one of the classical deep neural network models with excellent feature extraction ability, has become a research hotspot in the field of fault diagnosis [8]. Therefore, some researchers use the ability of CNN adaptive feature extraction to complete bearing fault diagnosis and get rid of the tedious manual feature extraction. Jing et al. [9] implemented fault diagnosis of planetary gearboxes by using a CNN to learn features directly from vibration signals, and experiments showed that the features extracted by convolutional

neural networks have higher diagnostic accuracy than artificial features. Wang et al. [10] used wavelet transform to convert rolling bearing vibration signals into two-dimensional images as input and then established a multi-scale feature fusion module (FMCNN) to extract different level features of fault samples, and the experimental results showed that the model can extract the features of each level of the signal and has high diagnostic accuracy and robustness under different noise interference. However, in the actual working environment, vibration signals are often interspersed with environmental noise, which can negatively affect the performance of the model when transformed into images along with the signal. Zhang et al. [11] took the raw vibration signal as the input to a wide convolutional neural network and used the wide convolutional kernel to enhance the resistance of the 1D convolutional neural network to noise, achieving a high diagnostic performance in a noisy environment. Gao et al. [12] proposed a novel hybrid deep learning method based on extended deep convolutional neural networks with wide first-layer kernels (EWDCNN) and long short-term memory (LSTM) to improve the fault diagnosis accuracy of rotating machinery in complex environments. Han et al. [13] proposed a new diagnostic framework that combined a spatiotemporal pattern network (STPN) with a CNN and applied it to fault diagnosis of complex systems. It can be seen that CNN has powerful feature extraction ability and effective fault diagnosis ability. Jiang et al. [14] proposed a wind turbine gearbox fault diagnosis model based on a multi-scale convolutional neural network (MSCNN), which captures diagnostic information at different scales by convolutional kernels of different sizes, and the results show that the method greatly improves the feature learning ability and achieves better diagnostic performance. Hao et al. [15] used ResNet for bearing fault diagnosis and used the global average pooling layer (GAP) instead of the fully connected layer (FC) to solve the problem of the high computational load of ResNet, and the experiments showed that the improved algorithm had a reliable fault diagnosis rate and the training time was reduced. It is evident that the convolutional neural network has a powerful feature extraction capability and effective fault diagnosis. However, the above CNN diagnostic model does not consider the weights of feature mappings in different convolutional channels, which leads to a certain degree of feature redundancy.

Recently, attention mechanisms in computer vision have attracted particular attention from researchers due to their ability to achieve better network performance by selectively enhancing useful information and reducing feature redundancy [16]. Hu et al. [17] proposed a "Squeeze-and-Excitation" module that significantly improves the ability of deep learning models to learn image features. Wang et al. [18] used a new channel and space attention neural network for image denoising, which was experimentally verified to have better visual results and a higher peak signal-to-noise ratio (SNR). Therefore, this work considers the introduction of channel attention into the field of bearing fault diagnosis to enable traditional CNNs to focus on the learning of useful features.

Inspired by the above research, a new bearing fault diagnosis model channel attention CNN (CA-CNN) is proposed here for bearing fault diagnosis to improve the diagnostic performance of deep learning models in noisy environments. The proposed model uses the raw vibration signal as input, utilizes wide kernel convolution to extract the global information of the vibration signal, extracts periodic features of the vibration signal by narrow kernel convolution at different scales, and embeds channel attention at different network depths to selectively enhance the fault classification features and reduce the influence of environmental information such as noise on the model. Finally, utilizing the global average pooling layer instead of the fully connected layer to reduce the computational effort while preventing overfitting, the softmax layer is used to diagnose the extracted bearing fault features.

The rest of this article consists of the following: In Section 2, an introduction to convolutional neural networks and channel attention is presented, and the network structure of the CA-CNN-based intelligent fault diagnosis method is explained in detail. In Section 3, to evaluate the performance of our method against existing methods, experiments are con-

ducted under different load and noise environments, and a discussion of the experimental results is presented. Section 4, concludes the study and presents directions for future work.

## 2. Methodology

In the training process of the convolutional neural network-based model, the convolutional operation generates feature maps of multiple channels, which contain rich feature information. However, the collected vibration signals contain information such as environmental noise and rotational speed, the generated feature maps also contain a lot of information irrelevant to fault diagnosis and the feature maps of different channels have different degrees of recognition of fault features. As a result, not all feature maps can express the fault features well. If all the feature maps of all channels are considered, it will instead increase the influence of redundant information in the input signal, which is unfavorable to the training of the model.

Based on the above idea, CA-CNN introduces channel attention into convolutional neural network for bearing fault diagnosis. By assigning different weight values to each channel, the model is allowed to focus on learning the channel features that are closely related to the diagnostic identification information to improve the fault classification. Next, we introduce the theory related to the construction of CA-CNN and the detailed fault diagnosis method.

### 2.1. A Brief Introduction to CNN

In recent years, convolutional neural networks have been widely used in bearing fault diagnosis. Convolutional neural networks generally consist of a convolutional layer, a pooling layer, a fully connected layer, and an output layer. The input to the convolutional neural network in the field of bearing fault diagnosis can be expressed as $X = [x_1, x_2, \ldots, x_t]$, where $t$ denotes the length of the original vibration signal. By calculating the result of the convolution kernel with the input signal one can obtain the features:

$$C_j = x_i * w_j + b_j \tag{1}$$

where $*$ is the dot product operation, $w_j$ is the weight matrix, and $b_j$ is the bias.

The activation function enhances the nonlinear representation of the network for the input signal, making it easy to separate different fault classes and accelerating model convergence. Taking the ReLU activation function used in this model as an example, its expression is shown below:

$$f(x) = \left\{ \begin{array}{l} x, x > 0 \\ 0, x \leqslant 0 \end{array} \right. \tag{2}$$

In order to reduce the size of the feature dimension, the output features in the convolution layer are usually fed to the pooling layer. The max pooling layer is used to select important features and is one of the most commonly used pooling layers currently. The pooling process can be described as follows:

$$P_j = \max\{C_j\} \tag{3}$$

After the convolution and pooling layers, the features extracted in the previous layers are mapped to one-dimensional vectors by a fully concatenated layer without losing information. Finally, the output layer is used to diagnose the health of the bearings.

### 2.2. Channel Attention

The channel attention structure used in this model is shown in Figure 1. It consists of a global pooling layer, two convolutional layers, a ReLU activation function, and a sigmoid activation function. The global pooling layer uses average pooling, which serves to compress each feature map in space, while the convolution layer and activation function assigns different weight values to the feature maps of different channels. For the input feature set

$M = \{m_1, m_2, \cdots, m_i, \cdots, m_c\}$, where $m_i \in \mathbf{R}^{1 \times w}$ represents some feature graph of length $W$. The feature set $M$ is first changed into $Z \in \mathbf{R}^{1 \times c}$ by global average pooling, which is the compression of the space features of each channel. The formula of compression is described as follows:

$$z_i = \frac{1}{1 \times W} \sum_{k}^{W} m_i(k) \tag{4}$$

The input feature $M$ is compressed in space to $Z$, and then it is transformed into $Z^*$ by two convolutional layers and activation function. The formula of the translation is described as follows:

$$Z^* = \delta(F_2(\sigma(F_1(Z)))) \tag{5}$$

where $F_1$ and $F_2$ denote that the number of channels is 1, the size of convolution kernel is $1 \times 1$ for convolution operation, $\sigma$ is the ReLU activation function, $\delta$ is the sigmoid activation function, $Z^*$ represents the importance of each channel, and different weight values are assigned to each channel. $Z^*$ is finally multiplied by the input feature set $M$ to obtain the new feature set $G$. The formula of $G$ is described as follows:

$$G = \{z_1^* m_1, z_2^* m_2, \cdots, z_c^* m_c\} \tag{6}$$

In order to reduce the effect of network depth on subsequent convolutional layers and to preserve the original information, the idea of residual learning is used to introduce residual connections into the calculation of channel attention to increase the possibility of further optimization. The formula is described as follows:

$$U = G + M \tag{7}$$

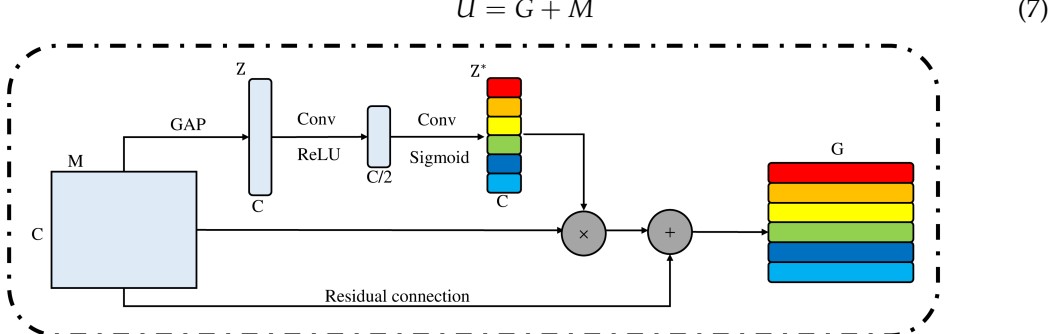

**Figure 1.** The structure of CA.

*2.3. Proposed CA-CNN Intelligent Diagnosis Method*

The structure of the CA-CNN model is similar to that of the traditional convolutional neural network, containing a convolution layer, pooling layer, and output layer, as shown in Figure 2. Considering that the traditional CNN method is limited by the size of the convolutional kernel when applied to vibration signals, a kernel that is too small may not be able to capture the global information of the vibration signal and thus is susceptible to local interference features. Therefore, CA-CNN employs a wide kernel in the first convolutional layer. To better capture the anomalous signals generated by the bearing at the failure point, borrowing from the successful application of Simonyan et al. [19] using multiple contiguous $3 \times 3$ narrow convolutional kernels in extracting local information in pictures, the remaining convolutional layers of CA-CNN use contiguous narrow kernels of different scales to extract the periodically varying features of the signal. Signals collected in industrial environments not only have bearing vibration signals, but they also contain some environmental noise unrelated to fault diagnosis. Therefore, in order to highlight the key fault information in the vibration signals and weaken the irrelevant environmental information, channel concerns are added after each convolutional layer, and different weights are assigned to the signal features of different channels. The global average pooling layer is used instead of the fully connected layer, so that no additional computational

parameters need to be introduced. This can be used to reduce the risk of model overfitting and improve the robustness to noise.

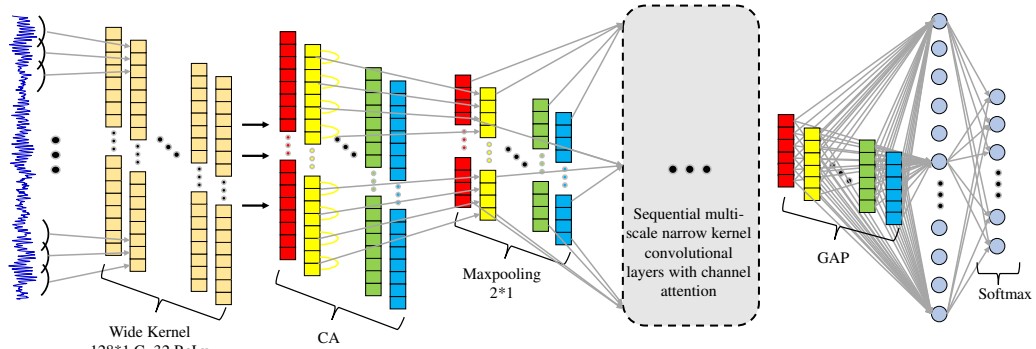

**Figure 2.** The structure of CA-CNN.

The input dimension of CA-CNN is (None, 1024, 1), which represents any number of raw vibration signals of length 1024 sample points. The vibration signal first passes through 32 wide kernels of $128 \times 1$ with a stride size of 8, which are used to capture the global information of the signal. Then, after the channel attention calculation to obtain the weighted expression of the feature dimension of (None, 128, 32). The feature dimension is then reduced by the max pooling layer of pooling size $2 \times 1$ to obtain an output of dimension (None, 64, 32). Then, by narrow convolutional layers at several different scales, it obtains a feature map of dimension (None, 14, 64). Next, the output dimension is (None, 64) through the global average pooling layer. Finally, the diagnostic results are output using softmax. Table 1 shows the detailed parameters of the network model.

**Table 1.** Detailed parameters of the CA-CNN.

| No | Layer | Kernel | Channel | Stride | Padding | Activation | Output |
|----|-------|--------|---------|--------|---------|------------|--------|
| 1 | Input | - | - | - | - | - | (None, 1024, 1) |
| 2 | Conv_1 | $128 \times 1$ | 32 | $8 \times 1$ | Yes | ReLU | (None, 128, 32) |
| 3 | CA_1 | - | - | - | - | - | (None, 128, 32) |
| 4 | MaxPooling_1 | $2 \times 1$ | - | $2 \times 1$ | No | - | (None, 64, 32) |
| 5 | Conv_2 | $9 \times 1$ | 64 | $1 \times 1$ | Yes | ReLU | (None, 64, 64) |
| 6 | CA_2 | - | - | - | - | - | (None, 64, 64) |
| 7 | MaxPooling_2 | $2 \times 1$ | - | $2 \times 1$ | No | - | (None, 32, 64) |
| 8 | Conv_3 | $6 \times 1$ | 64 | $1 \times 1$ | Yes | ReLU | (None, 32, 64) |
| 9 | CA_3 | - | - | - | - | - | (None, 32, 64) |
| 10 | MaxPooling_3 | $2 \times 1$ | - | $2 \times 1$ | No | - | (None, 16, 64) |
| 11 | Conv_4 | $3 \times 1$ | 64 | $1 \times 1$ | Yes | ReLU | (None, 16, 64) |
| 12 | GAP | - | - | - | - | - | (None, 64) |
| 13 | Softmax | - | 10 | - | - | Softmax | (None, 10) |

## 3. Results and Discussion

This section first introduces the experimental setup and dataset information for bearing faults based on vibration signals. Next, the performance of CA-CNN for bearing fault diagnosis under different loads is evaluated using the bearing dataset. Then, the fault diagnosis performance of CA-CNN in noisy environments is evaluated by constructing noisy datasets and compared with other fault diagnosis methods. After that, the model training process is compared and analyzed in order to demonstrate the effect of channel attention on the noise immunity of the model. Finally, the overall performance of the model for all classes is measured by confusion matrix and t-SNE feature visualization.

### 3.1. Data Description

In the experiment, the rolling bearing dataset used was provided by the Bearing Data Center at Case Western Reserve University (CWRU) [20]. The test rig is schematically shown in Figure 3 and consists of a motor drive end bearing, a torque transducer, and a power tester. The test rig uses a 16-channel data logger with a sampling frequency of 12 kHz to acquire vibration signals and a torque transducer to measure load and speed.

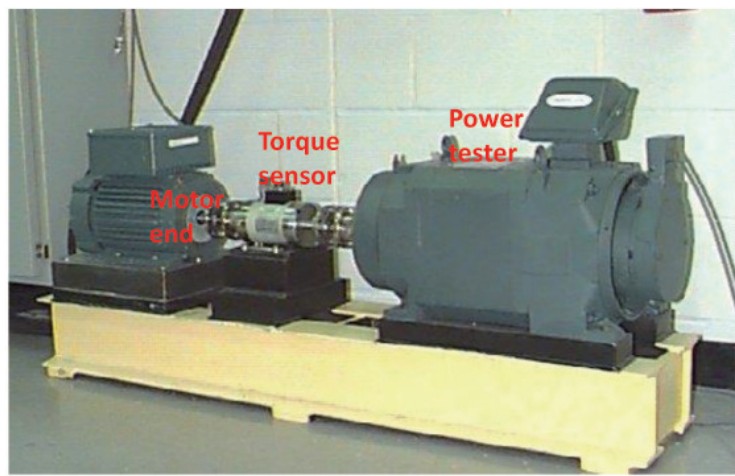

**Figure 3.** Schematic diagram of CWRU rolling bearing test platform.

The experiments were conducted using vibration signals collected from rolling bearings of motor drive end model SKF6205 at loads of 0 HP, 1 HP, 2 HP, and 3 HP. In order to make the collected faulty data as realistic and valid as possible, faulty bearings were obtained by machining single point damage using an EDM machine. The data contain four different statuses: normal (N), rolling ball faulty (BF), inner ring faulty (IF), and outer ring faulty (OF). Each faulty status has three sizes of damage: 0.18 mm, 0.36 mm, and 0.54 mm, respectively. In total, there are 10 bearing statuses for each load. To ensure the rich feature information of the experimental samples, each sample length is set to 1024 sample points in the experiment, where each load has 2000 samples for the training set, 1000 samples for the validation set, and 1000 samples for the test set. The definition of the labels for the different bearing statuses is shown in Table 2. The raw vibration signals of the 10 bearing statuses at 0HP are shown in Figure 4.

**Table 2.** Definition of bearing status labels.

| Status | Labels | | | | | | | | | |
|---|---|---|---|---|---|---|---|---|---|---|
| | **0** | **1** | **2** | **3** | **4** | **5** | **6** | **7** | **8** | **9** |
| Location | BF | BF | BF | IF | IF | IF | OF | OF | OF | N |
| Size (mm) | 0.18 | 0.36 | 0.54 | 0.18 | 0.36 | 0.54 | 0.18 | 0.36 | 0.54 | - |

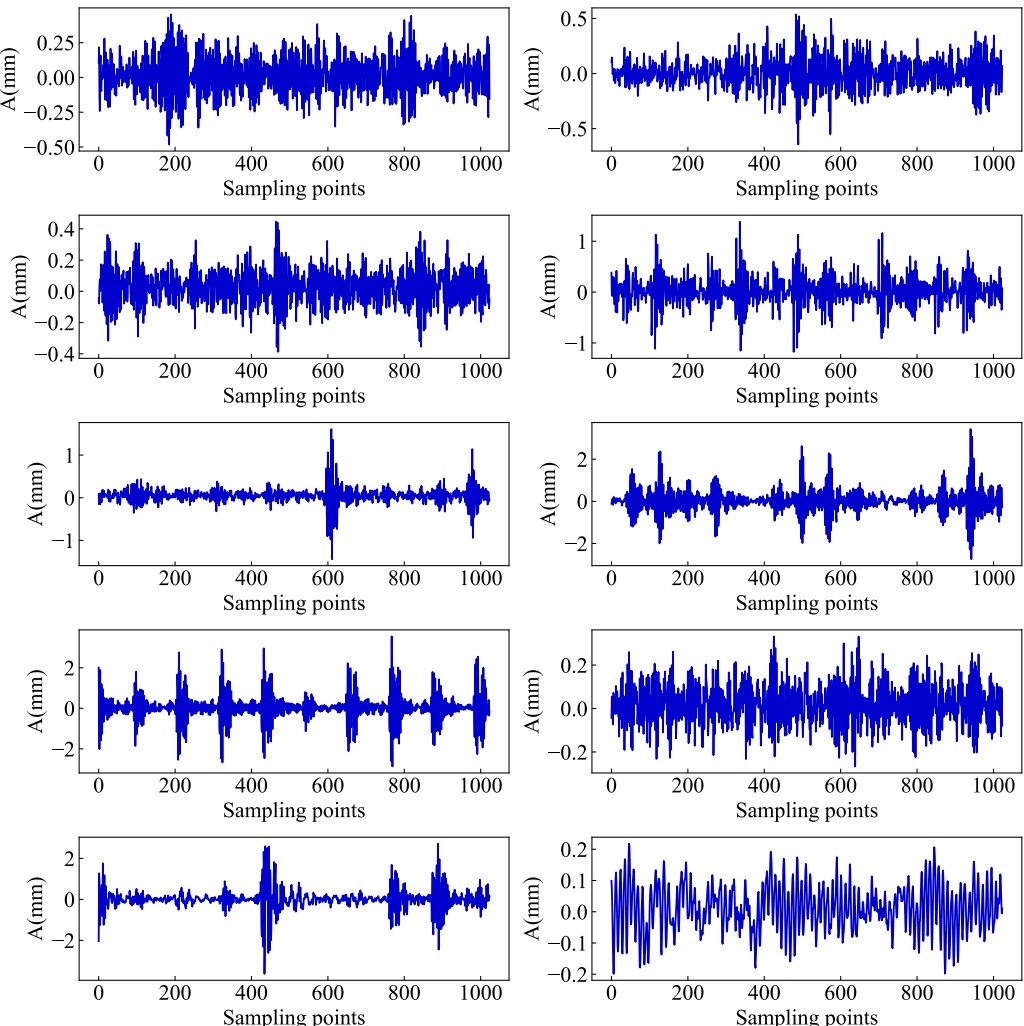

**Figure 4.** Raw vibration signals of 10 bearing status at 0HP. The *y*-axis represents the amplitude of the signal, and the *x*-axis represents the number of samples.

### 3.2. Parameters Setting

Experiments were conducted with Windows 11 operating system, AMD R7 4800H CPU, RTX2060 graphics card, and experimental code written mainly in the Python language on the Tensorflow 2.0 framework. The model batch size was set to 64, Adam was used as the optimization algorithm, and the learning rate was set to 0.001.

The major difference between CA-CNN and standard CNN is that CA-CNN embeds channel attention at different network depths. For the requirement of subsequent experiments, the network structure of the standard CNN is given in Table 3.

**Table 3.** Detailed network structure of the standard CNN.

| No | Layer | Kernel | Channel | Stride | Padding | Activation |
|----|-------|--------|---------|--------|---------|------------|
| 1 | Conv_1 | $128 \times 1$ | 32 | $8 \times 1$ | Yes | ReLU |
| 2 | MaxPooling_1 | $2 \times 1$ | - | $2 \times 1$ | No | - |
| 3 | Conv_2 | $9 \times 1$ | 64 | $1 \times 1$ | Yes | ReLU |
| 4 | MaxPooling_2 | $2 \times 1$ | - | $2 \times 1$ | No | - |
| 5 | Conv_3 | $6 \times 1$ | 64 | $1 \times 1$ | Yes | ReLU |
| 6 | MaxPooling_3 | $2 \times 1$ | - | $2 \times 1$ | No | - |
| 7 | Conv_4 | $3 \times 1$ | 64 | $1 \times 1$ | Yes | ReLU |
| 8 | GAP | - | - | - | - | - |
| 9 | Softmax | - | 10 | - | - | Softmax |

*3.3. Fault Diagnosis Results under Different Load*

In order to verify the effectiveness of the proposed model, a total of six different models using machine learning-based RF, SVM, and deep learning-based 1D-CNN [21], MA-CNN [22], WPE-CNN [23], and standard CNN (CNN) were tested for comparison to demonstrate the effectiveness of CA-CNN in bearing fault diagnosis. To ensure the accuracy of the test results, we used the 10-fold cross-validation method to reduce the chance of random assignment of the training and test sets by dividing the raw vibration data into 10 equal-sized subsets (among them, 9 subsets were used as the training set and 1 subset was used to test the model), and the cross-validation was repeated 10 times. Each subset was used only once as validation), and the final result is the average of the 10 cross-validations. In addition, we added a judgment statement to the CA-CNN to stop the iteration when the training set data had iterated 10 times consecutively and the loss value was less than 0.15 at the same time to speed up the diagnosis. Table 4 shows the diagnostic accuracy of each model under different loads, and the diagnostic accuracy of CA-CNN under different loads is 99.9%, 99.2%, 99.9%, and 99.9%, respectively. The diagnostic accuracy of CA-CNN under 0HP is 24.9% and 21% higher than that of RF and SVM, respectively. The results show that CA-CNN can achieve an end-to-end diagnosis of bearing vibration signals. It can also be seen that the performance of the machine learning-based bearing fault diagnosis method is inferior to that of the deep learning-based bearing fault diagnosis method.

**Table 4.** The diagnostic accuracy of models under different loads.

| Model | 0 HP (%) | 1 HP (%) | 2 HP (%) | 3 HP (%) | Avg (%) |
|---|---|---|---|---|---|
| RF | 0.750 | 0.767 | 0.790 | 0.817 | 0.781 |
| SVM | 0.789 | 0.685 | 0.709 | 0.762 | 0.736 |
| 1D-CNN | 0.993 | 0.976 | 0.995 | 0.999 | 0.991 |
| MA-CNN | 0.994 | 0.994 | 0.998 | 0.999 | 0.997 |
| WPE-CNN | 0.988 | 0.988 | 0.994 | 0.994 | 0.991 |
| CNN | 0.978 | 0.987 | 0.990 | 0.980 | 0.984 |
| CA-CNN | 0.999 | 0.992 | 0.999 | 0.999 | 0.998 |

Figure 5 visualizes the diagnostic accuracy and the average diagnostic accuracy of the models under different loads. The proposed model has an average accuracy of 99.8%, which is higher than the other six models. The reason for this is that the proposed model combines multi-scale convolution and channel attention mechanisms, which can automatically extract bearing fault features and increase the weights of effective features to assist the fault classifier in achieving a higher diagnostic accuracy. These results demonstrate the effectiveness of CA-CNN in fault diagnosis tasks.

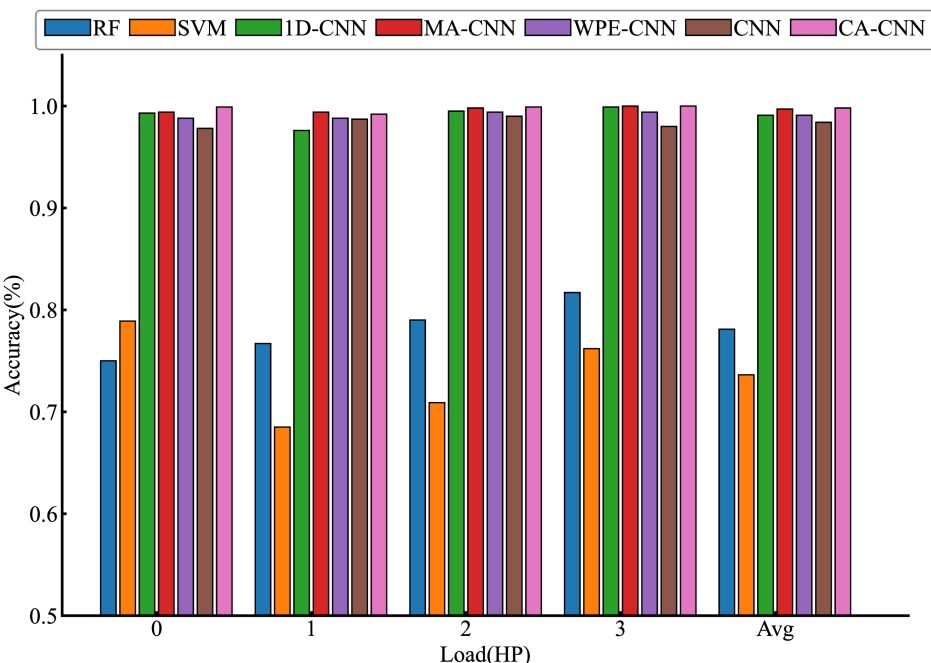

**Figure 5.** Diagnostic accuracy of models under different loads.

### 3.4. Fault Diagnosis Results under Noise Environment

In the actual working environment, the collected monitoring data inevitably contains environmental noise information. To validate the diagnostic performance of the proposed model under noise interference, this study added Gaussian noise with SNR of −4∼8 dB to the raw signal to simulate the environmental noise and make the test dataset more closely match the actual working environment. The SNR calculation equation is described as follows:

$$\text{SNR} = 10\lg\left(\frac{P_s}{P_n}\right) \tag{8}$$

where $P_s$ is the energy of the signal and $P_n$ is the energy of the noise.

According to the above formula, it is known that the smaller the SNR, the greater the noise energy. At SNR = 0 dB, the noise energy is one times the signal energy. Figure 6 shows the visualization plots of the raw samples and the samples when interfered with SNR = −2 dB noise. The fluctuating characteristics of the vibration signal are almost covered by the noise, which will certainly have a negative impact on the models' performance.

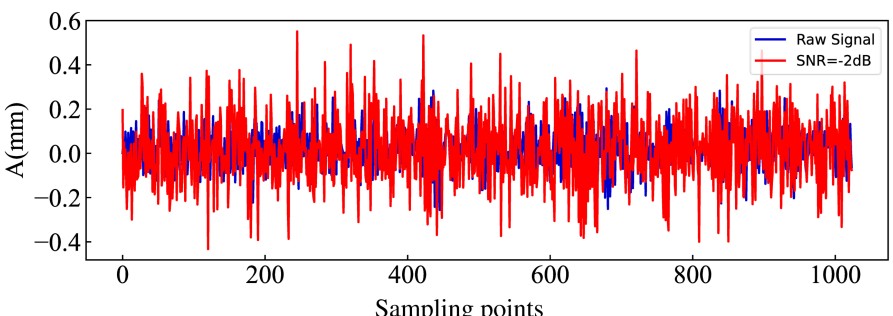

**Figure 6.** Visualization of the raw signal and the signal after adding −2 dB noise.

To verify the diagnostic ability of the proposed model in a noisy environment, it is compared to four models, FMCNN [10], WDCNN [11], 1D-CNN [21], and standard CNN(CNN). The diagnostic results of each model in the noise interference environment are shown in Table 5 and Figure 7.

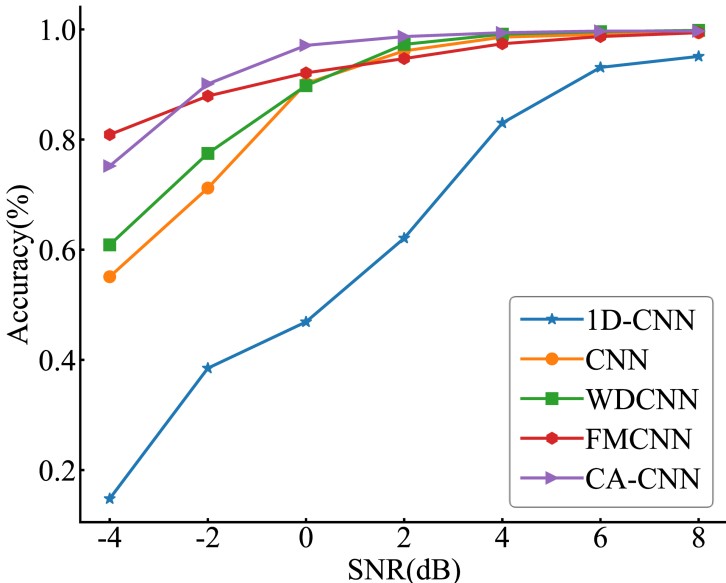

**Figure 7.** Accuracy of models under different noise environments.

**Table 5.** Accuracy of models under different noise environments.

|  | FMCNN | WDCNN | 1D-CNN | CNN | CA-CNN |
|---|---|---|---|---|---|
| Highest accuracy (%) | 0.994 | 0.998 | 0.951 | 0.994 | 0.997 |
| Lowest accuracy (%) | 0.809 | 0.609 | 0.148 | 0.551 | 0.752 |
| Average (%) | 0.930 | 0.898 | 0.619 | 0.871 | 0.943 |
| Time (s) | 158 | 54 | 1333 | 59 | 86 |

Table 5 shows that CA-CNN nearly performs the best with different SNRs. The average diagnostic accuracy is 94.3%, which is better than other models. The minimum diagnostic accuracy is higher than WDCNN (89.8%), 1D-CNN (14.8%), and CNN (55.1%) and the maximum diagnostic accuracy is better than 1D-CNN (95.1%), FMCNN (99.4%), and CNN (99.4%). CNN performs worse than CA-CNN in noisy environments, which is due to the fact that CA-CNN uses channel attention to weaken the interference of noise on feature extraction and make the model more focused on the fault features. It can also be seen in Table 5 that the average training time in CA-CNN is 86s, which satisfies our requirement for training time. The 1D-CNN has the worst performance in a noisy environment with low diagnostic accuracy and high training cost due to the overfitting of the model caused by the deep network structure of the 1D-CNN. The other CNN models took less time to train and did not exhibit overfitting in the noisy environment because they used relatively shallow network structures. By comparison, CA-CNN has a more comprehensive performance.

In Figure 7, it is clear that the accuracy of each model increases as the SNR increases, and the change in accuracy smooths out for most models after SNR = 2 dB. The accuracy of CA-CNN is higher than most models at SNRs of −4 dB to −2 dB. The accuracy of CA-CNN reached nearly 99% at SNR = 2 dB. This indicates that CA-CNN can effectively perform fault diagnosis in noisy environments without any data anti-noise processing and is robust in noisy environments.

### 3.5. Training Process Comparison

Because the structure of CNN and CA-CNN models are more similar, in order to analyze the training situation and performance of the models, CNN and CA-CNN are selected for comparison in a strong noise environment, and both have the same optimizer, batch size, and learning rate. Figure 8 shows how the accuracy of CNN and CA-CNN changes during the training process. From Figure 8, it can be seen that the accuracy of CNN and CA-CNN are almost the same at the beginning of training, which is due to the

fact that the channel attention used by CA-CNN is still at the stage of feature selection and because the residual connection retains the original information of the signal, so the performance of CNN and CA-CNN is similar at the beginning of training. At a number of iterations of 40, CA-CNN has completed convergence, and the accuracy no longer changes significantly. As the iteration proceeds, channel attention gradually suppresses redundant features and focuses on key local features, which greatly enhances the learning ability of CA-CNN. Furthermore, the use of a global pooling layer instead of a fully connected layer is an important factor for the fast and stable training speed of CA-CNN.

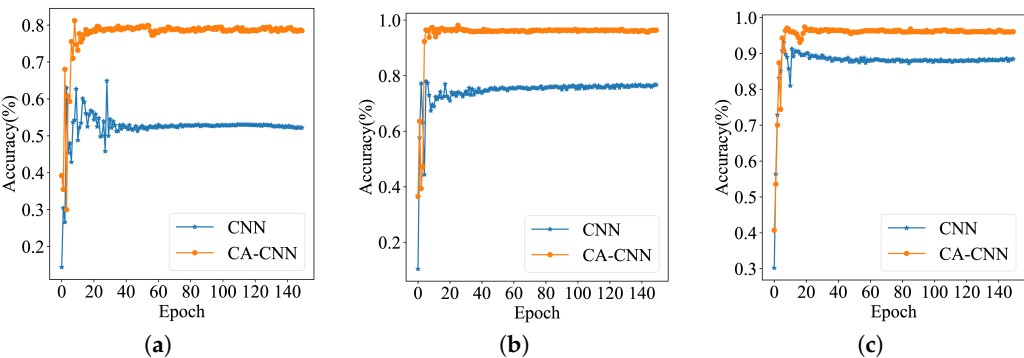

**Figure 8.** Training process comparison. (**a**) SNR = −4 dB. (**b**) SNR = −2 dB. (**c**) SNR = 0 dB.

*3.6. Visual Analysis of Diagnosis Results*

In order to analyze and compare the classification accuracy, precision, and recall of CA-CNN and CNN for each bearing status in different noise environments, we use the confusion matrix to summarize the results in Figures 9 and 10. The confusion matrix is a result analysis table used to evaluate the performance of the classifier. The coordinate axes represent the bearing status, each of its columns represents the predicted label, each row represents the true label, and the value represents the accuracy of the classification. The confusion matrix from different SNR environments shows that the two models have different degrees of misclassification for the fault statues of rolling ball faulty (0.36 mm/0.54 mm) and inner ring faulty (0.36 mm/0.54 mm), and it is not difficult to infer that this is due to the negative effect of noise on the performance of the CNNs, which makes the class mismatch of the faulty status with similar vibration signals. In the absence of adding channel attention, this effect is particularly significant. With added channel attention, the identification accuracy of the bearing status of normal, rolling ball faulty (0.18 mm), inner ring faulty (0.18 mm), and outer ring faulty (0.18 mm/0.36 mm/0.54 mm) is almost 100% in the environment of SNR = −4 dB. It is worth noticing that in the environment with SNR = −2 dB, 20% of the rolling ball faulty (0.54 mm) is misclassified as a rolling ball faulty (0.18 mm) by CA-CNN. In the environment of SNR = −4 dB, 40% of the rolling ball faulty (0.54 mm) is misclassified as rolling ball faulty (0.18 mm) and 10% is misclassified as inner ring faulty (0.18 mm). This indicates that strong noise very easily masks the faulty information of the rolling ball faulty class (0.54 mm), which leads to confusion about the model for similar faulty types.

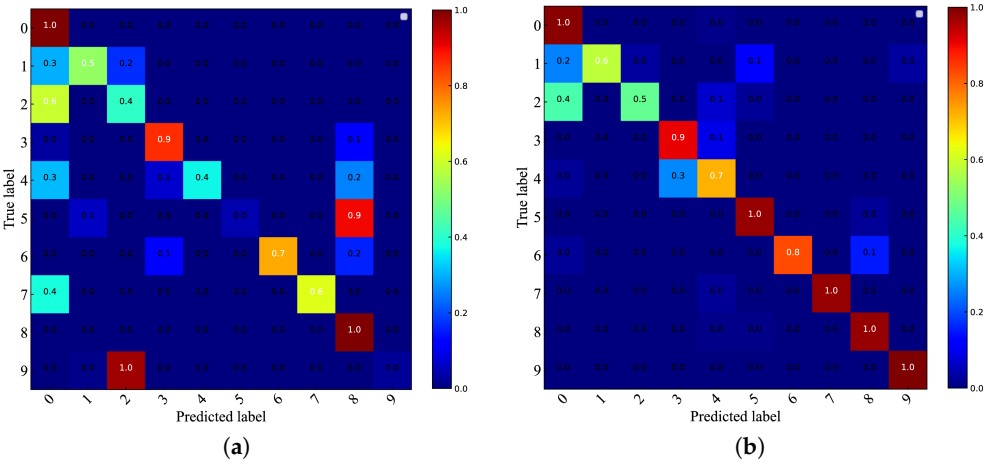

**Figure 9.** Confusion matrix at SNR = −4 dB. (**a**) CNN. (**b**) CA-CNN.

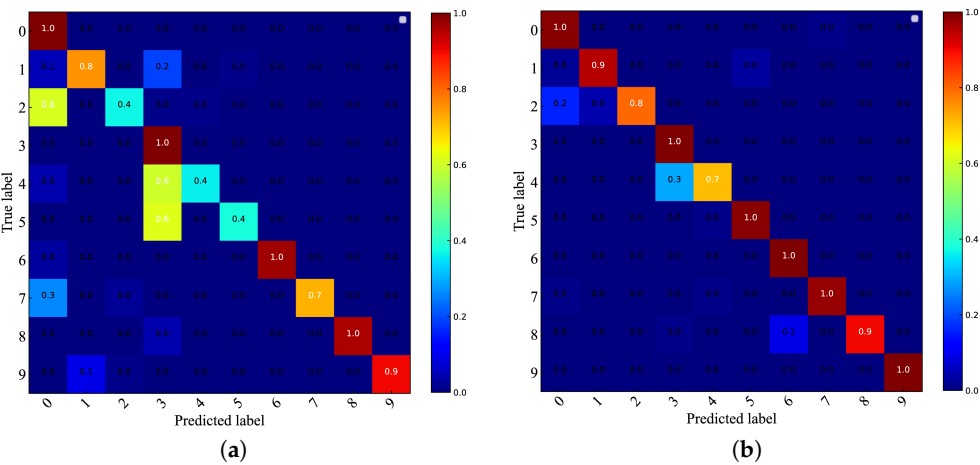

**Figure 10.** Confusion matrix at SNR = −2 dB. (**a**) CNN. (**b**) CA-CNN.

### 3.7. Visual Analysis of Features

To intuitively demonstrate the feature extraction capability of each module of the model species in a noise environment, the features extracted from each convolutional and output layer of the proposed model at SNR = 0 dB are visualized using t-SNE [24] to observe the feature separation results of different convolutional layers in noise environment. t-SNE is a dimensionality reduction tool for visualizing high-dimensional datasets in two or three dimensions in a lower dimensional space. Its horizontal and vertical coordinates represent the two dimensions, and the values represent the relative distance between discrete points. Visualization results of t-SNE features at SNR = 0 dB noise environment are shown in Figure 11. To facilitate the demonstration of the feature aggregation capability of CA-CNN, we visualize the data distribution of vibration signals in the first plot, different colors and numbers represent the faulty status of the bearing, and the detailed faulty statuses can be found in Table 2 with corresponding information. The features with the same faulty status are clustered together, while the features with different faulty statuses are separated from each other under the action of t-SNE. As observed in Figure 11, in Conv_1, the features of the other fault classes are surrounded by an inner ring faulty (0.54 mm) and overlap. However, after the channel attention and Conv_2, the various faulty types start to cluster. It is easy to notice that the outer ring faulty class (0.18 mm) is the easiest to segment and is already separated from the other classes at Conv_2. As the convolutional layers of different scales with added channel attention go deeper, the discrete points gradually

disappear, indicating that the proposed model is able to distinguish the faulty features in a noisy environment.

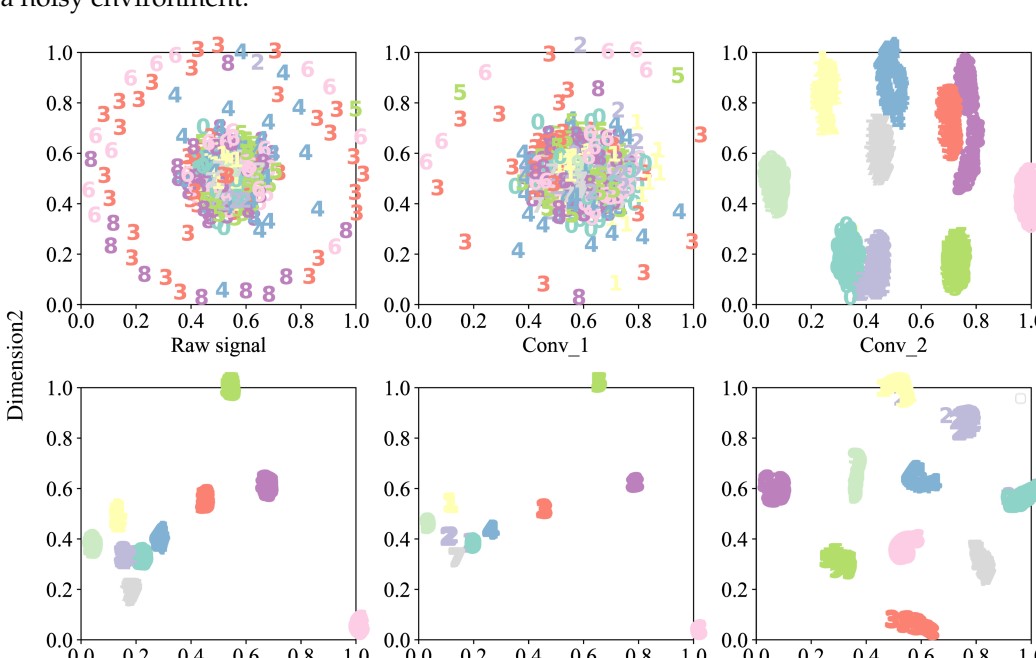

**Figure 11.** Features visualization by t-SNE at SNR = 0 dB.

## 4. Conclusions

In this paper, the current research on bearing fault diagnosis is systematically described. A new bearing fault diagnosis model, CA-CNN, is proposed to introduce channel attention to the fault diagnosis of bearing. The global information of the signal is captured using wide kernel convolution, the periodic variation information of the signal is extracted using narrow kernel convolution at different scales, the effectiveness of the convolutional neural network in identifying fault features is improved by adaptively weakening the interference of environmental noise on the raw vibration signal using channel attention, and the number of training parameters is greatly reduced by using a global pooling layer instead of a fully connected layer. The model was validated using rolling bearing vibration data provided by Case Western Reserve University. A series of experiments yielded the conclusions that the proposed model has higher diagnostic accuracy, faster convergence, and greater resistance to noise than existing models. More details will be confirmed in future studies, which will include more extended datasets.

**Author Contributions:** Conceptualization, J.Y. and G.C.; methodology, J.Y. and G.C.; formal analysis, J.Y. and G.C.; writing—original draft preparation, J.Y. and G.C.; writing—review and editing, J.Y. and G.C.; visualization, G.C. and J.Y. All authors have read and agreed to the published version of the manuscript.

**Funding:** This research was funded by the Humanity and Social Science General (Planning Fund) of the Ministry of Education of China (Grant Nos.17YJA880004).

**Data Availability Statement:** Data openly available in a public repository. The data that support the findings of this study are openly available in Case Western Reserve University Bearing Data Center at https://engineering.case.edu/bearingdatacenter/apparatus-and-procedures.

**Conflicts of Interest:** The authors declare no conflict of interest. The funders had no role in the design of the study; in the collection, analyses, or interpretation of data; in the writing of the manuscript; or in the decision to publish the results.

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
