# Peer review of "Intelligent Motor Bearing Fault Diagnosis Using Channel Attention-Based CNN"

_wevj, doi:10.3390/wevj13110208_

Round 1

Reviewer 1 Report

The paper presents the application of a CA-CNN to the diagnosis of faults in motor bearings. The originality of the paper is not very high, because in literature there are already many papers dealing with the topic of fault diagnosis for motor bearing based on CNNs. Moreover, the authors use the CWRU database which has been widley used, while the use of different databases and hence measurements would improve the research in this field.

However, the authors propose a slightly different CNN architecture (the CA-CNN) which could be worth to be published. It is very important the comparison of the CA-CNN performance with that of the standard CNN: in view of this, the CNN structure must be better described. Please, provide a full description of the standard CNN, so the reader can understand the differences in the layers between the two CNNs.

In Fig. 9, the accuracy of the two CNNs is shown: from the plot, it seems that the standard CNN has not converged yet. The referee recommends to run the CNN training with more epochs, to be sure that the training arrives at convergence.

The visual analysis of the features could be very interesting, but, for a reader who is not familiar with this tool, it is not easy to understand. Please, try to better explain the images in Fig. 12: which is the meaning of the numbers in the first plot? Which is the meaning and the measurement units of the axes?

Minor: in page 3, line 99 in the sentence "the result of convolution  of the convolution kernel..." maybe should be modified in "the result of the convolution kernel..."

Reviewer 2 Report

This is an interesting paper about bearing fault diagnosis. A new bearing fault diagnosis model, CA-CNN, is proposed to introduce chan- 338 nel attention to the fault diagnosis of bearing. A series of 346 experiments yielded that the proposed model has higher diagnostic accuracy, faster 347 convergence, and greater resistance to noise than existing models.

Thus, I recommend to accept this paper for the publication in the journal of World Electr. Veh..

Reviewer 3 Report

- I found some typos, please do general proofreading of the text.

- In the introduction, the direct quote style is confusing, make a refit.

- For better style, every section beginning should contain a preamble.

- The spacing between equations and sections is small.

- Tables 1, 2, and 3 are outside the margins.

- In Section 3.1 it was not clear how the bearing failures were produced.

- In Section 3.3 it was not clear how the tests were carried out to the point of not overfitting.

- Regarding the computational cost, how do CNNs behave?

Round 2

Reviewer 1 Report

The authors have improved the manuscript, following the referee's suggestions. Now it can be published in the present form.